# Molecular Mechanisms of Diverse Auxin Responses during Plant Growth and Development

**DOI:** 10.3390/ijms232012495

**Published:** 2022-10-18

**Authors:** Yang Zhang, Jiajie Yu, Xiuyue Xu, Ruiqi Wang, Yingying Liu, Shan Huang, Hairong Wei, Zhigang Wei

**Affiliations:** 1Engineering Research Center of Agricultural Microbiology Technology, Ministry of Education, Heilongjiang University, Harbin 150500, China; 2State Key Laboratory of Tree Genetics and Breeding, Northeast Forestry University, Harbin 150040, China; 3College of Forest Resources and Environmental Science, Michigan Technological University, Houghton, MI 49931, USA; 4Heilongjiang Provincial Key Laboratory of Plant Genetic Engineering and Biological Fermentation Engineering for Cold Region, School of Life Sciences, Heilongjiang University, Harbin 150080, China

**Keywords:** auxin, auxin response mechanism, transcription, growth and development, protein interaction, phosphorylation

## Abstract

The plant hormone auxin acts as a signaling molecule to regulate numerous developmental processes throughout all stages of plant growth. Understanding how auxin regulates various physiological and developmental processes has been a hot topic and an intriguing field. Recent studies have unveiled more molecular details into how diverse auxin responses function in every aspect of plant growth and development. In this review, we systematically summarized and classified the molecular mechanisms of diverse auxin responses, and comprehensively elaborated the characteristics and multilevel regulation mechanisms of the canonical transcriptional auxin response. On this basis, we described the characteristics and differences between different auxin responses. We also presented some auxin response genes that have been genetically modified in plant species and how their changes impact various traits of interest. Finally, we summarized some important aspects and unsolved questions of auxin responses that need to be focused on or addressed in future research. This review will help to gain an overall understanding of and some insights into the diverse molecular mechanisms of auxin responses in plant growth and development that are instrumental in harnessing genetic resources in molecular breeding of extant plant species.

## 1. Introduction

The signaling mediated by auxin, a major phytohormone, acts as a master regulator in virtually every aspect of plant growth and development, such as embryogenesis, apical dominance, lateral root formation, hypocotyl elongation, and tropic responses to light and gravity [1]. Extensive studies have suggested that auxin signaling can be split into three aspects: biosynthesis and metabolism [2,3], directional transport [4], and cell/tissue-specific responses [5,6]. Because cell/tissue-specific auxin response is often the final and decisive step of auxin functions in regulating plant growth and development [7], unveiling the underlying molecular mechanisms of the diverse auxin responses has become one of the hotspots in plant research [8,9,10,11,12,13]. To date, a great progress has been made in elucidating the underlying molecular mechanisms of the diverse auxin responses [11,14,15,16,17,18,19,20,21,22,23,24,25], and there are many reviews about auxin responses [2,12,14,16,26,27,28,29,30,31]. However, most reviews only provide a summary of molecular characteristics of a specific auxin response, rather than a panoramic view of the auxin responses and the divergence between the diverse auxin responses. Owing to the importance of auxin in almost every aspect of plant growth and development, it is imperative to systematically summarize and compare the characteristics of the diverse auxin responses, which is instrumental for comprehending the overall molecular mechanisms and gaining a holistic view of auxin functions in plant growth and development. According to whether auxin responses are involved in transcriptional changes, this review divides them into transcriptional auxin response (TAR) and non-transcriptional auxin response (Non-TAR) pathways, and then narrates the current models and recapitulates key interactive and the regulatory importance of these two types of auxin responses. In addition, given that the transcription regulation plays vital roles, not only in plant growth and development, but also in plant response to environmental cues, we emphasize the coupled mechanisms of various TARs. This review is conductive to comprehending the molecular mechanisms underlying diverse auxin responses and conducting molecular breeding of plants.

## 2. Transcriptional Auxin Response

### 2.1. The Paradigm of the Canonical Transcriptional Auxin Response (CTAR)

Auxin can cause multiple transcriptional reprogramming in various cell types, tissues, and whole plants, which is known as TAR [30]. Among these TARs, the CTAR includes three canonical core components, auxin/indole-3-acetic acid (Aux/IAA) proteins, SCF (Skp-Cullin-F-box) ubiquitin ligase (E3) complex containing transport inhibitor response 1 protein (TIR1) or auxin signaling F-box protein(AFB) (SCF^TIR1/AFB^), and auxin response transcription factors (ARFs), which have been widely studied and thoroughly reviewed in previous studies [11,26,31]. Here, we primarily recapitulated the molecular mechanism of CTAR (Figure 1). In the absence or at low levels of auxin, even though ARFs can bind to auxin response elements (AuxREs) present in their target gene promoters, their transcriptional activities are inhibited by dimerization with Aux/IAA proteins through their Phox and Bem1 (PB1) domains. Subsequently, the Aux/IAAs interact with corepressors of TOPLESS (TPL) or TPL-RELATED (TPR) proteins through their ethylene response factor-associated amphiphilic repression domains (EAR), and the C-terminal double WD40 motifs of TPLs/TPRs, which recruits histone deacetylase (HDAC) complexes and chromatin-modifying proteins, thereby allowing AUX/IAA to mediate nearby chromatin condensation and transcriptional suppression [32]. When auxin increases and is perceived by SCF^TIR1/AFB^ complex, it acts as a “molecular glue” to promote interactions between TIR1/AFB and Aux/IAA proteins. Aux/IAA proteins are then ubiquitinated and recognized by 26S proteasome-mediated degradation machinery. The degradation of Aux/IAA co-repressors leads to the unfolding of condensed chromatin and releases ARFs from inhibition by Aux/IAA, allowing them to activate the expression of auxin-responsive genes [12,15,33]. Although the three core components and molecular mechanism of CTAR appear to be uncomplicated [26], the fact that the three components belong to multigene families functioning in different cell/tissue types and are subjected to multilevel regulations can diversify and specialize the outputs of CTAR, offering an explanation of how such a simple molecular mechanism can regulate a wide array of growth and developmental processes in various cell/tissue types across numerous species. For this reason, CTAR is also an important focus of the auxin molecular biology.

#### 2.1.1. Transcriptional Property of ARFs in CTAR

The ARF contains a non-conserved region referred to as the middle region (MR), which has been proposed to function as either a transcriptional repression or activation domain [34]. The ARFs with Q-rich MRs are considered to function as activators, whereas most, if not all other ARFs, function as repressors. It has been reported that the ARFs involved in the CTAR are only the matter of activator ARFs [10]. In contrast, repressor ARFs function via squelching the transcriptional activity of activator ARFs either by heterodimerizing with them to compete for DNA binding sites or recruiting transcriptional co-repressor such as TPL or TPL-related proteins to hamper transcription [35,36,37]. Actually, three mechanisms adopted by the repressor ARFs in auxin response may exist concurrently during plant growth and development [30,38]. In addition, the activator and repressor ARFs may be co-expressed in the same cell/tissue to establish an equilibrium that is presumably more robust than an auxin response that relies only on single-activator or single-repressor ARFs [9,35]. For example, repressor ARF1 and ARF2 share redundant functions with activator ARF6 in auxin-dependent cell division of embryo [39]. The balanced levels of repressor and activator ARFs can keep a constant transcriptional response even in the presence of fluctuating auxin signals, based on their extensive co-expression [35]. In addition, it is worth noting that most ARFs in plants are repressors, with 17 out of 23 ARFs being repressors in *Arabidopsis* [40], indicating that the elaborate mechanisms employing repressor ARFs have been evolved and functioned in plants, and therefore attention needs to be paid to this in future research.

#### 2.1.2. Characteristics of CTAR

#####  Rapidity

Because the Aux/IAA degradation mediated by auxin is very rapid [41], the rapid transcriptional reprogramming is a salient feature of CTAR [15] (Figure 1). Accordingly, it has been reported that the half-lives of AUX/IAAs are as short as 6 to 8 min [42], and the expression of most *AUX/IAA* genes is rapidly (<15 min) induced by auxin [41]. As the result of the rapidity, the dynamic spatio-temporal changes in auxin levels can provoke transcriptional reprogramming rapidly. For example, some auxin-induced transcripts were observed to quickly increase within 5 min upon auxin treatment [43,44]. The rapidity of CTAR enables quick and abrupt changes of various physiological and metabolic processes that are essential for plants growth and development as well as survival under various erratic stress environments as being sessile organisms.

##### Diversity and Specificity

Besides rapidity, the outputs of CTAR manifest drastic specificity and diversity [45] (Figure 1). The mechanisms underlying this scenario mainly include the following aspects. First, the spatiotemporally divergent expressions of TIR1/AFB, Aux/IAA, and ARF family comprising multiple members contribute to different large combinations of protein interactions with very different biochemical affinities [46,47,48], which lays the main molecular basis for the diversity and specificity of CTAR [24]. Second, among auxin-responsive genes, some are auxin-induced or -repressed secondary transcriptional factors (TFs), which, accompanied by their tissue-specific co-factors, can further render specificity and diversity of CTAR [49]. Third, the core components of CTAR are subjected to multilevel regulations, which in turn tremendously enhances specificity and diversity of CTAR [29,38,50]. Fourth, most members of Aux/IAA and ARF families have three domains, through which they interact with other auxin signaling components, and act as tuning knobs to specify and define the output properties for a given auxin signal [7]. In addition, the sequence variation within these domains may furthermore increase interaction specificity between core components, and concurrently enhance specificity and diversity of CTAR [24]. Fifth, besides TGTCTC AuxRE [27], the ARFs also bind to other AuxRE types such as TGTCNN [48,51]. In addition, these AuxREs may occur in the form of direct or palindromic repeats in the promoters [52]. Moreover, as the high-efficient binding of ARFs to AuxREs needs at least two sites, sequence specificity and orientation between the same and distinct AuxREs have great impacts on the binding affinities of ARFs [24]. Furthermore, ARF monomers do not bind to AuxREs as efficiently as ARF dimers, especially when binding to palindromic AuxREs [53], and the dimerization mediated by dimerization domain rather than PB1 domain of some ARFs [48], such as ARF1 and ARF5 [48], preferentially binds to AuxREs with different spacings (also named “molecular calipers”) [24,48]. Finally, the aforementioned mechanisms act alone or together to guarantee the diversity and specificity of CTAR in a context dependent manner. In summary, owing to many factors that affect the diversity and specificity of CTAR, it generates dynamics and specificity to vast transcriptional outputs to meet various needs for plant growth and development and adaptation to environmental cues.

#### 2.1.3. Multilevel Regulations of the CTAR

As described previously [2,24,38,45], the three core components of CTAR are subjected to multilevel regulations (Figure 1), such as (1) transcriptional regulation through negative feedback loop via auxin, alternative transcript splicing [24,54,55]; (2) post-transcriptional regulation through miRNA and transacting-small interfering RNA [30,56,57]. For example, stress-induced miRNA393 targets *TIR1/AFB* mRNAs for degradation, while auxin-induced miRNA847 accumulation decreases the abundance of *IAA28* transcripts [58]; (3) post-translational regulation through S-nitrosylation, phosphorylation, and ubiquitination [16,59,60,61]; (4) regulation mediated by protein interactions such as TIR1 interactions with HEAT SHOCK FACTOR 90 and SUPPRESSOR OF G2 ALLELE SKP1, agonistically interaction between RGA-LIKE3 and IAA17, ARF6 interaction with BZR1 and PIF4 [24,62,63,64]; *Aux/IAA* transcription is also regulated by abiotic stress-induced DREB/CBF TFs and the light-regulated PIF4/5 and HY5 TFs [58]; (5) regulation mediated by oligomerization of the three core components such as TIR1-TIR1, Aux/IAA-Aux/IAA, and ARF-ARF [24,65]; (6) regulation mediated by nucleo-cytoplasmic partitioning and condensation formation regulation [66,67,68]. For example, the nucleo-cytoplasmic partitioning of ARF19 leads to a shift in the transcriptional landscape of auxin response [68]. In addition, ARF5 is capable of unlocking packed chromatin in concert with the SWI/SNF chromatin remodelers BRHAMA (BRM) and SPLAYED (SYD) [69], and GRE motif-binding bZIP transcription factors can recruit the histone acetyltransferase SAGA complex to a *GH3* gene and induce auxin-responsive transcription [70]. These aforementioned multilevel regulations not only guarantee the accuracy of CTAR outputs, but also reflect that a variety of plant developmental factors and environmental cue-enabled factors participate in CTAR, which in turn enhances diversity, specificity, and complexity of CTAR in a context-dependent manner (Figure 1).

### 2.2. Non-CTAR

Besides CTAR, several other auxin responses involved in transcriptional reprogramming, also referred to as TAR, function during some plant growth and development processes and responses to environmental stresses.

#### 2.2.1. TAR Mediated by Remodeling Chromatin via MONOPTEROS/ARF5

The transcriptional reprogramming via specific chromatin region remodeling in plants has been reported to be executed by a complex containing switching defective/sucrose nonfermenting (SWI/SNF) chromatin remodeling ATPases such as BRM and SYD in *Arabidopsis* [71], which functions to unlock the condensed packaged chromatin for transcriptional activation [72]. Consistent with this, MONOPTEROS (MP)/ARF5 interacts with a complex comprising BRM or SYD via its middle domain [73], leading to transcriptional alternations via remodeling specific chromatin regions instead of auxin-induced degradation of Aux/IAA proteins as CTAR does in the condition of high auxin [69]. In the absence of, or at a low level of auxin, the expressions of auxin response genes are repressed by MP through interacting with its partner IAA12/BODENLOS (BDL), which together with TPL, recruits HDA19 to maintain chromatin at the target loci in an unlicensed and repressive state (Figure 2A). In addition, the Aux/IAA-TPL complex non-competitively inhibits the SWI/SNF complex comprising BRM or SYD to associate with MP, which also inhibits transcriptional reprogramming in an auxin-dependent manner (Figure 2A). When auxin levels increase, the SWI/SNF complex containing BRM or SYD and the BSH subunit is physically associated with MP after BDL proteolysis via the TIR1/AFB pathway, which facilitates a permissive chromatin conformation and consequently makes the AuxREs, formerly occluded by nucleosomes, available for this transcriptional complexes and co-regulation at the promoters of auxin responsive genes [9] (Figure 2A). This TAR mediated by MP mainly acts during cell-fate reprogramming in the tissues such as embryogenesis, root development, seedling viability, and leaf development [69]. In addition, there are additional proteins involved in this TAR, such as chromatin remodeling component PROPORZ1 in histone acetylation [74], PLETHORA [75], and PHD-finger protein complexes containing OBERON1 and TITANIA [76], which also affect chromatin states at MP target loci in embryonic root meristem initiation [9].

The MP may auto-regulate its own transcription, as well as that of its repressor, IAA12/BDL, to thus generate two feedback loops regulated by auxin, demonstrating that MP can serve as a ‘switch’ for promoting IAA12/BDL degradation when auxin is above a threshold concentration [9], which restitutes the original Aux/IAA protein abundance and reverts the system to its ground state after an auxin pulse [5,77]. In addition, auxin-mediated degradation of Aux/IAAs activates MP and then directly up-regulates about one-half of *Aux/IAA* genes in both roots and shoots of *Arabidopsis*, which then in turn, represses MP activation [78].

Compared with CTAR, the TAR mediated by MP preferentially functions in the specific tissues with rapid dividing cells, where its interaction complexes comprising BRM and SYD are present and needed for rapid transcriptome reprogramming. In addition, the transcriptional reprogramming via MP activation is more immediate and rapid compared with that mediated by CTAR because the switching between repressive and de-repressive chromatin states in an instantly-reversible manner is aided by SWI/SNF complex in TAR mediated by MP whereas CTAR has no specific protein complex to promote depolymerization of chromatin from repressive state.

#### 2.2.2. TAR Mediated by SCF SKP2A-E2F/DPB

The cell cycle is a pivotal process and is under the control of strict regulatory systems. E2-promoter binding factor C (E2FC) interacts with dimerization partner B (DPB) by form heterodimer, which represses S-phase genes of cell-cycle in differentiation processes of plants [80,81], and can be degraded by ubiquitin–proteasome system (UPS)-dependent proteolysis and thus alleviates their repression of cell cycle genes [80,82,83]. In the absence of or at a low level of auxin, the heterodimers, comprising E2FC and DPB, repress cell-cycle genes that contain the E2F binding sites in their promoters [79]. The E2Fc has been shown to interact with DPB in its nonphosphorylated form; when E2Fc is phosphorylated, the formation of the E2Fc/DPB heterodimer is lost (Figure 2B). When auxin levels increase, S-phase kinase-associated protein 2 (SKP2A), a F-box protein and a core component of the SCF^SKP2A^ complex with E3 ubiquitin ligase activity like SCF^TIR1/AFB^ in CTAR, promotes degradation of heterodimers comprising E2FC and DPB repressors through directly binding to auxin, and thereafter E2F and DP positive dimers, including E2FA or E2FB [84], bind to E2F sites in the promoters of cell-cycle genes and activate their transcription [82,83], which forms a part of the G1/S checkpoint in cell-cycle progression where some TFs and proteins need to be degraded before the next phase commences [80] (Figure 2B). Afterwards, E2FC repressors again occupy these E2F binding sites to limit the re-entry into a new cell cycle, accordingly preventing a non-programmed cell division [25] (Figure 2B). In addition, SKP2A binding to auxin also contributes to its own proteolysis and then enhances degradation of E2FC and DPB again [25], which not only prevents SKP2A over-function but also establishes transcriptional regulatory loops, accompanying with the UPS-dependent proteolysis, to ensure a balanced level of proteins needed in each phase of the cell cycle [82].

Although CTAR mediated by the SCF^TIR1/AFB^ complex, accompanied by TAR mediated by the SCF^SKP2A^ complex, participates in the regulation of the cell cycle [82], the genes regulated by the SCF^SKP2A^ pathway are merely confined to cell cycle process, whereas the genes in CTAR also involve other biological processes required for plant normal growth and development and adaption to adverse environmental conditions. In addition, TAR mediated by the SCF^SKP2A^ pathway regulates the cell cycle more accurately and quickly as it specifically degrades E2FC and DPB repressors and activates E2F and DP positive dimers, which especially bind to the E2F sites in the promoters of cell-cycle genes, instead of AuxREs.

#### 2.2.3. TAR Mediated by Non-Canonical ARF or Aux/IAA Proteins

Besides canonical ARF and Aux/IAA proteins, there are also some ARF and Aux/IAA proteins that do not have some of the four typic domains but are known to participate in TARs in some specific biology processes as auxin level increases. The TARs mediated by these atypical ARF and Aux/IAA proteins mainly include the following two categories.

#### TAR Mediated by ETT/ARF3 without PB1 Domain

Among the ARF family of *Arabidopsis*, ARF3, ARF13, and ARF17 do not have PB1 domains and thus do not interact with Aux/IAA proteins as the typic ARFs do [9,85]. Unlike ARF13 and ARF17 which are simply truncated ARFs [9], ARF3, also referred to as ETTIN (ETT), possesses a specific C-terminal domain (ETT-specific, ES domain) [86], which resembles the DNA binding domain of ARF4 and promotes itself interactions with TFs in an auxin-dependent mechanism that requires neither ubiquitination nor TIR1 [17]. Hence, ETT adopts an alternative mechanism to translate auxin signal into transcriptional outputs without involving protein degradations, which is different from what canonical activator ARFs do in CTAR [17,87] (Figure 2C). In the absence of or at a low level of auxin, ETT interacts with TPL/TPR2 at target loci via EAR motif within its ES domain, and then recruits HDA19 for histone (H3K27) deacetylation to maintain chromatin in a condensed state, which represses the expression of ETT target genes [28,88]. When auxin increases, its direct binding to the ETT breaks the interaction between ETT and TPL/TPR2, and HDA19 disassociates from ETT, thus enhancing H3K27 acetylation, which relieves chromatin condensation and then activates the expressions of ETT target genes [17,28]. Compared with CTAR that is dependent on resynthesis of Aux/IAAs for resetting repression in the condition of high auxin level, the TAR mediated by ETT has an advantage in the speed as it allows switching between repressive and de-repressive chromatin states in an instantly-reversible manner, which is especially important for developmental switches such as gynoecium morphogenesis [17,28].

Although ETT should principally function as a repressor ARF based on the amino acid composition of its middle region [40], it actually acts as both a transcriptional repressor and an activator, which depends not only on interacted TFs, such as floral homeotic AGAMOUS and KANADI1 [89,90], but also auxin levels in diverse developmental contexts [87]. In addition, ETT functions as a central node in coordinating auxin dynamics and plant development because it interacts with diverse proteins partners and then regulates various genes involved in transcriptional regulation, multiple hormone dynamics, and plant organ development in an auxin-sensitive manner [17,87]. In summary, TAR mediated by ETT plays a major role in some specific developmental processes, such as gynoecium development and patterning [87,91], that need the expression of genes to be transformed rapidly from their transcriptional activation to repression states in an instantly-reversible manner.

#### TAR Mediated by Non-Canonical Aux/IAA Proteins

Among 29 AUX/IAA proteins in *Arabidopsis*, there are six AUX/IAA proteins that do not have some domains compared with canonical AUX/IAA proteins [92]. Of these six genes, IAA32, IAA33, and IAA34 have been reported to participate in TARs [16,60].

(1) TAR mediated by untypical IAA33

Gene IAA33 has no typical domains I and II, which determine the interactions between AUX/IAA and TIR1 proteins and are essential to repress ARF-mediated transcription in CTAR [16,24]. Hence, IAA33 regulates auxin-dependent gene expressions through competing with canonical AUX/IAA proteins such as IAA5 to interacts with repressor ARF10/16 rather than activator ARFs in specific biological processes such as root stem cell identity [16,93,94] (Figure 2D). In the absence of or at a low level of auxin, although IAA33 is degraded by a 26S proteasome [16], IAA5 binds to repressor ARF10/16, which releases their repression activity for *WUSCHEL-RELATED HOMEOBOX 5* (*WOX5*), a major regulator of the root stem cell activity [94]. Then, the transcription of *WOX5* is activated and thus keeps root stem cell identity via inhibiting root stem cell differentiation [94]. In addition, IAA17/AXR3 also up-regulates *WOX5* expression via currently unknown mechanism, and further inhibits root stem cell differentiation [94]. When the auxin levels increase, auxin induces the phosphorylation activity of MITOGEN-ACTIVATED PROTEIN KINASE 14 (MPK14), which phosphorylates IAA33 and, thus prevent it from being degraded by 26S proteasome machinery. The phosphorylated IAA33 interacts with repressor ARF10/16 [16,95], which derepresses the transcriptional repression of *WOX5* and keeps identity of some root stem cells. On the contrary, the IAA5 and IAA17/AXR3 are degraded via the SCF^TIR1/AFB^ pathway [16], and do not interact with ARF10/16 anymore. These repressor ARF10/16 inhibit the expression of *WOX5*, which leads to most root stem cell differentiation [16,94].

Notably, unlike the expression of canonical *AUX/IAAs*, such as *IAA5* and *IAA17/AXR3* that are regulated at both transcriptional and translational levels in an auxin-dependent manner [46], the expression of *IAA33* is not affected by auxin treatment [16]. In addition, in order to prevent an excessive transcriptional response in a specific biology process like stem cell identity in roots with high level auxin, TAR mediated by MPK14-IAA33-ARF10/16 module operates in parallel with CTAR mediated by IAA5 and IAA17/AXR3 degradation [16], which generates diverse transcriptional responses for accurate regulation of root stem cell identity.

(2) TAR mediated by untypical IAA32/IAA34

Although IAA32/IAA34 do not have typical domains I and II as IAA33 does [92], they are phosphorylated by the transmembrane receptor kinase 1 (TMK1) rather than MPK14 at the high level of auxin at the concave side of the apical hook in *Arabidopsis* [16,60]. The phosphorylated IAA32/34 increase their stability and then maintain repression of ARFs such as ARF2 and ARF7, while the canonical AUX/IAA proteins are degraded through the SCF^TIR1/AFB^ pathway, leading to the release repression of ARFs [60], which represents a distinct TAR that is somewhat similar with TAR mediated by the IAA33 phosphorylation but different with CTAR mediated by the AUX/IAAs degradation [16,44] (Figure 2E). Notably, the TAR mediated by TMK1-IAA32/IAA34-ARF2/7 model has been so far reported to only regulate concave side development of the apical hook in *Arabidopsis* [60]; whether it participates in other biological processes in the tissues with high level auxin needs to be studied further. In any circumstance, this may indicate that there are potentially more TAR pathways that resemble those mentioned above but may function spatiotemporally within the plants.

(3) Differences between TARs mediated by IAA33 and IAA32/34 proteins

Although both IAA33 and IAA32/34 repress the auxin-response gene expression through their phosphorylation to inhibit growth in the tissues with high level of auxin, there are some differences between them. First, IAA32/34 binds to not only activator ARFs but also repressor ARFs [60], whereas IAA33 preferentially interacts with repressor ARFs [16]. Second, IAA32/IAA34 are regulated not only at post-translational level through phosphorylation mediated by TMK1 but also at transcriptional level through CTAR [60], whereas IAA33 is only regulated at phosphorylation level mediated by MPK14 [16]. Third, the TAR mediated by the TMK1-IAA32/34-ARF2/7 model functions at the concave side of the apical hook, whereas the TAR mediated by the MPK14-IAA33-ARF10/16 model participates in root stem cell identity [16,60].

In summary, TARs mediated by both the MPK14-IAA33-ARF10/16 and TMK1-IAA32/34-ARF10/16 models act as repressors for some auxin-response genes accompanying with activation of other auxin-response genes through CTAR in the developmental tissues with high auxin, which prevents an excessive transcriptional response and contributes to fine-tuning transcriptional response in a tissue-specific manner. However, whether these two TAR pathways participate in other biological processes needs to be studied further.

## 3. Auxin Response Mediated by Auxin Binding Protein 1 (ABP1)

Besides transcriptional reprogramming, auxin also triggers some specific responses such as microtubule organization [96], clathrin-mediated endocytic trafficking [97], and intracellular trafficking of PIN-FORMED (PIN) auxin efflux transporters [98], which occur too quickly to be a result of transcriptional remodeling or de novo protein synthesis [99]. The ABP1, an endoplasmic reticulum-targeting protein, has been proposed to play a vital role in these Non-TARs [97,99]. However, there is also one study that shows ABP1 is not essential for auxin response in plant normal growth and development [100], which is contrary to most previous researches about ABP1’s functions in an auxin-dependent growth and development and needs to be studied further due to challenge in characterizing such an embryo lethality protein. To date, ABP1 is still considered to be an important mediator of an auxin-dependent growth and development as well as abiotic and biotic stresses. For example, conditional repression of *ABP1* reveals its important roles in coordinating cell division and cell expansion during postembryonic shoot development in both *Arabidopsis* and tobacco (*Nicotiana benthamiana*) [101]. Decreased ABP1 activity led to severe retardation of leaf growth, an altered cell plates and shapes, and a decrease in cell expansion [101]. Here, we recapitulate the two main mechanisms of ABP1 participating in auxin response during plant growth and development.

### 3.1. Non-TAR Mediated by ABP1

When ABP1, located at the plasma membrane (PM) [102,103], binds to auxin, it interacts with PM-localized transmembrane proteins such as TMK1, SPIKE1 (SPK1), or other unidentified proteins, which transmit auxin signal from extracellular space into cells to activate critical molecular switches, Rho of plants (ROP)-GTPases (ROPs) [104,105]. The activated ROPs subsequently interact with their effectors such as CRIB motif-containing proteins (RICs) and thereby trigger multiple Non-TARs in the cytosol [104,106]. Based on the proteins that directly interact with ABP1, the Non-TARs can be mediated by the following two pathways.

#### 3.1.1. Non-TAR Mediated by ABP1-TMK1-ROPs

The Non-TAR mediated by the ABP1-TMK1-ROPs pathway plays a critical role in the spatial control of cell division and expansion during plant growth, morphogenesis, and development, which can be illustrated by the leaf pavement cell development in *Arabidopsis*. Then, ABP1, once perceiving auxin, activates TMK1 auto-phosphorylation, and then triggers two counteracting ROP2-RIC4 and ROP6-RIC1 signaling pathways [106,107,108] (Figure 3A). The ROP2-RIC4 pathway induces accumulation of cortical F-actin and then promotes lobe outgrowth [109,110], whereas ROP6-RIC1 promotes ordered cortical microtubule (MT) polymerization through activation of MT-severing protein katanin at the indenting regions, and thereby inhibits indentation outgrowth [111,112]. Meanwhile, ROP2 inactivates RIC1 which in turn suppresses ROP2 activation in indentation zones; this contributes to counteractivity of outgrowth-promoting ROP2 and outgrowth-inhibiting RIC1 pathways to coordinate interdigitation growths between adjacent pavement cells [113].

The antagonistic interaction between the ROP2-RIC4 and the ROP6-RIC1 pathways has been also shown to regulate PIN1 polar transport across the PM, through endocytosis/exocytosis during leaf pavement cell development in an auxin-dependent manner [97,104,105,110] (Figure 3A). The PIN1 endocytosis occurs at the indentation regions but its internalization into the endosomal compartments is inhibited at the lobe regions, where the ROP2-RIC4 pathway promotes the assembly of fine cortical microfilaments and thus inhibits PIN1 endocytosis [109]. Meanwhile, PIN1 transports intracellular auxin outside to PM, which in turn activates ROP2 through the ABP1-TMK1 pathway to construct a positive feedback loop regulation for the PIN1 and auxin polar distribution [106,109]. In addition, PIN1-exported auxin at the lobe sites can diffuse across the cell wall to coordinately activate the ROP6-RIC1 pathway at the complementary indenting site [114].

#### 3.1.2. Non-TAR Mediated by ABP1-SPK1-ROPs

Besides TMK1 in the ABP1-TMK1-ROP signaling pathway, SPK1 in the ABP1-SPK1-ROP singling pathway generates activated ROPs via its catalysis when interacting with auxin-binding ABP1 [115], which subsequently activates effectors such as RICs to regulate the Non-TAR pathway [116] (Figure 3B). For example, once perceiving the auxin signal translated by ABP1 in roots, SPK1 positively regulates the ROPs-RICs pathways and the Wiskot–Aldrich syndrome protein-family Verprolin homology protein (WAVE) and actin nucleation/branching by the Actin-Related Protein2/3 (Arp2/3) heteromeric complexes, which thereby activate actin polymerization and inhibit PINs endocytosis to promote polarized growth of root cells [115,117,118]. The cortical actin networks, which are stabilized not only by the ROP6 effector protein RIC1 but also via WAVE and the Arp2/3 heteromeric complexes, directly regulate polarized growth of root cells [115,119,120]. In addition, the ROP6-RIC1 pathway inhibits PIN2 internalization and thereby promotes PIN2 retention at the PM of the apical ends of epidermal cells and the basal ends of cortical cells in the root tips [105], which takes part in a connected circulatory auxin flow that is downward in the stele or cortex and upward in the epidermal cells of roots [121,122].

#### 3.1.3. Differences between the Non-TARs Mediated by ABP1-TMK1-ROPs and ABP1-SPK1-ROPs

Although both of the two Non-TARs mediated by the ABP1-TMK1 and ABP1-SPK1 models regulate polar growth through changing cytoskeleton based on microtubule and actin movement and regulating the PINs’ polar distribution on PM to maintain auxin uneven diffusion outside PM, there are some differences between them in auxin response. First, the Non-TAR regulates leaf pavement cell development only through the ABP1-TMK1 model for polar growth in the differential zone, whereas it modulates root polar growth through the ABP1-SPK1 model and WAVE and the Arp2/3 heteromeric complexes. Second, the Non-TAR mediated by the ABP1-TMK1 model maintains polar growth via PIN1 in leaf pavement cell development, while the Non-TAR mediated by the ABP1-SPK1 model utilizes PIN2 to keep polar growth in root cells. Third, the Non-TAR mediated by the ABP1-TMK1 model needs low auxin concentrations in leaf pavement cell development, whereas higher auxin concentrations are required for triggering Non-TAR mediated by the ABP1-SPK1 model in roots.

### 3.2. TAR Mediated by ABP1

Besides being involved in diverse Non-TARs as mentioned above, ABP1 has been proposed to take part in transcriptional reprogramming in an auxin-dependent manner [101,123]. For example, in the *ABP1* knockdown, the steady-state level and auxin sensitivity of transcripts of some early auxin response genes, such as *Aux/IAAs*, *SAUR*, and *GH3*, decrease [101,124]. In addition, inactivation of *ABP1* not only alters the expression of core cell cycle genes related to the G1/S checkpoint and the G2/M transition, but also significantly decreases or inhibits the expression of the mitotic markers of CYCB1D-box genes in the shoot apical meristems and root apical meristems [125,126]. Additionally, inactivation of *ABP1* causes significant changes in the expression of genes involved in cell wall remodeling [127].

Up to now, ABP1 has been suggested to participate in regulating expression of these aforementioned genes possibly through the following mechanisms (Figure 3C). First, given that ABP1 either acts on some TFs such as MYB77, which was reported to interact with ARFs (e.g., ARF7) in the TAR pathway [128], or by one means or another alters the relative affinity interactions between Aux/IAA and ARF proteins, which governs the expression of auxin response genes, and act as negative regulator at starting point of CTAR [129], ABP1 regulates gene expression either in concert with CTAR or through acting antagonistically to CTAR [127]. Second, besides cytoplasmic functions, ROPs not only affect the expression of auxin response genes in their overexpression tobacco and *Arabidopsis* transgenic lines [130], but also stimulate the formation of nuclear protein bodies that contain components of the TIR1–AUX/IAA ubiquitination pathway in CTAR [131]. Hence, ABP1 may participate in an auxin-dependent transcriptional reprogramming via indirect activation of ROPs through direct acting on TMK1 or SPK1. Third, multiple signaling components such as MAP kinases [95], IBR5 protein phosphatase [132], and phospholipase A2 [133], which have been reported to be involved in auxin response but have not been implicated in CTAR, are putative candidates mediating ABP1 action on regulating the expression of auxin response genes [123]. However, how these regulators interact with each other to mediate ABP1 to regulate an auxin-dependent gene expression remains to be determined and needs to be studied further. Finally, it is worth noting that no matter which aforementioned mechanism ABP1 adapts to in TAR, it should indirectly regulate the expression of auxin-dependent genes as it sits at the PM and in the endoplasmic reticulum, whereas the transcriptional reprogramming takes place in the nucleus.

## 4. Genetic Modification of Plant Growth by Diverse Auxin Response Genes

As aforementioned, auxin is a key regulator of virtually every aspect of plant growth and development. Therefore, genetic modification of genes related to diverse auxin responses should generate great effects on various traits of interest. To examine this, we recapitulate some genes involved in diverse auxin responses that were genetically modified in plants. As shown in Table 1, many complex traits of interest are altered, indicating that the modification of these genes can generate the traits as we expected, which lays the basis for precise editing using the recently developed new technologies. First, one dissatisfaction is that the up/down-regulation of some genes related to diverse auxin responses may cause the changes of multiple traits in multiple tissues, which can be circumvented by using the tissue-specific promoters. Second, recently developed technologies, for example, the clustered regularly interspaced short palindromic repeats (CRISPR)-Cas (CRISPR-associated proteins)-based editing has provided revolutionary tools for gene knockouts, base editing, prime editing, homology-directed repair, targeting gene activation/silencing, and epigenomic editing [134]. These technologies enable the modification of the DNA-binding domains and protein interaction domain of TFs. We can adapt the CRISPR/dCas9-act gene activation system [135] to enhance some gene expression and CRISPR/Cas13 gene silencing system that uses sgRNAs target message RNAs to reduce gene expression [136].Third, we can use recombinase technology, which is a flexible and effective system for stably stacking multiple genes within an *Agrobacterium* virulence plasmid transfer DNA. With this method, it is possible to assemble and deliver a 28 kilobase (kb) T-DNA with 10 cargo sequences into *Arabidopsis* [137] and *Solanum tuberosum* [138] and a 37 kb, 11-stack T-DNA into *Oryza sativa* [139], which allows us to modify multiple genes related to diverse auxin responses to manipulate multiple traits with higher efficacy in one effort. Finally, some regulatory mechanisms in diverse auxin responses can be used to construct the synthetic promoters or regulatory systems. For example, the positioning of two or more AuxREs motifs with 5–9 bps enables dimerization of ARFs, which can amplify the auxin response [48]. In addition, the orientation of these AuxREs can define the auxin response. As we have shown above, many genes involved in diverse auxin responses have not been genetically engineered with aforementioned tools or methodologies, and we believe these tools and strategies will allow us to understand auxin responses and their underlying molecular mechanisms better and acquisition of expected traits and better performance for economic benefit and environmental adaptation.

## 5. Conclusions and Future Perspectives

The phytohormone auxin participates in almost all the growth and development processes, as well as responses to the various environment cues in terrestrial plants. A key question in plant auxin biology is to explain how this simple generic signaling molecule can trigger such an enormous range of very specific outputs and seemingly omnipotence effects [8,10]. In the last few decades, dramatic progress on auxin biology has uncovered the fundamental molecular mechanism of how auxin is perceived and how its signal is converted to unique and specific responses. According to whether auxin response involves in transcriptional reprogramming, it can be divided into TAR and Non-TAR. In addition, TAR includes two pathways, CTAR and Non-CTAR, based on the structural characteristics of their core components. Although CTAR appears to be short and simple, it generates numerous transcriptional outputs with diversity and specificity, which is caused primarily by multilevel regulations, and fulfill most needs of various biological processes of plant growth and development in an auxin-dependent manner. Meanwhile, the Non-CTAR and Non-TAR only participle in some specific biological processes such as embryonic root meristem initiation and interdigitation growth between adjacent pavement cells of leaves in *Arabidopsis*, respectively.

Despite intensive efforts in auxin response research, many central questions are yet to be addressed: (1) the biological functions and regulation mechanisms of Non-CTAR and Non-TAR need to be further studied; (2) the elucidation of how the TAR and Non-TAR pathways are integrated, especially how they are linked/bridged by other plant signal transduction pathways, will advance our understanding and foster the utilization of the knowledge in real application; (3) the potential effects of specific auxin responses at different concentrations remains unexplored; (4) it is still unclear how repressor ARFs regulate gene repression or whether their functions involve auxin responses; (5) although we now have a better understanding of the activation of gene expression by auxin sensation and perception, how auxin represses genes is still unknown. Future research should unveil if the auxin sensation and perception processes use the same components that activate gene expression in the presence of auxin, or these processes involve other unknown proteins; (6) how diverse auxin responses crosstalk with other hormones-dependent response pathways for fine-tuning the growth and differentiation. Understanding of all these questions will allow us to fathom and decipher the sophisticated regulation and eventually pave a path towards utilization.

Finally, genetic modification of genes involved in diverse auxin responses manifests the alternations of plant architectures, stem characteristics and forms, leaf, flower, and fruit development, as well as response to various stresses. Therefore, genetic modification of these genes through traditional gene manipulation and precise editing and fine-tuning of these genes with recently developed CRISPR technologies hold great promise for generating new varieties and cultivars of vegetables, fruits, and staple crops to enrich our food, and ornamental plants with desired traits to decorate our landscapes and surroundings.

## Figures and Tables

**Figure 1 ijms-23-12495-f001:**
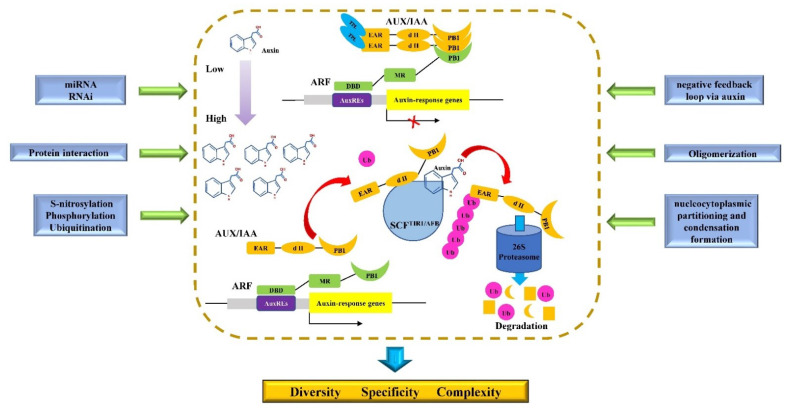
Regulation model of the canonical transcriptional auxin response. AUX/IAA: auxin/indole-3-acetic acid; ARF: auxin response transcription factor; TPL: TOPLESS; EAR: ethylene response factor-associated amphiphilic repression domains; PB1: Phox and Bem1; DBD: B3-type DNA binding domain; SCF^TIR1/AFB^: SCF (Skp-Cullin-F-box) ubiquitin ligase (E3) complex containing transport inhibitor response 1 protein (TIR1) or auxin signaling F-box protein (AFB); Ub: ubiquitin; AuxREs: auxin response elements (Adapted with permission from Ref. [31]. 2016, Li).

**Figure 2 ijms-23-12495-f002:**
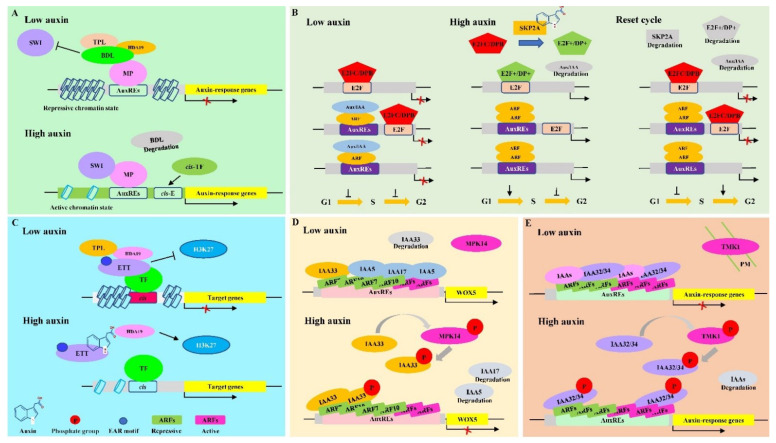
Regulation model of the non-canonical transcriptional auxin response. (**A**). Regulation model of transcriptional auxin response (TAR) mediated by remodeling chromatin via MONOPTEROS/ARF5. TPL: TOPLESS; SWI: switching defective/sucrose nonfermenting; BDL: BODENLOS; MP: MONOPTEROS; TF: transcriptional factor; HDA19: histone deacetylase 19 (Adapted with permission from Ref. [69]. 2015. 2015, Wu); (**B**). Regulation model of TAR mediated by SCF^SKP2A^ -E2F/DPB. E2FC/DPB: E2-promoter binding factor C (E2FC) interacts with dimerization partner B (DPB) to form heterodimers; AUX/IAA: auxin/indole-3-acetic acid; ARF: auxin response transcription factor; SKP2A: S-phase kinase-associated protein 2; E2F^+^/DP^+^: active E2F and DP positive dimers; G1: gap 1 phase of mitosis; S: synthesis phase of mitosis; G2: gap 2 phase of mitosis (Adapted with permission from Ref. [79]. 2014, Del Pozo); (**C**). Regulation model of TAR mediated by ETT/ARF3 without PB1 domain. ETT: ETTIN; Blue oval: ethylene response factor-associated amphiphilic repression domains (EAR motif) (Adapted with permission from Ref. [17]. 2016, Simonini); (**D**). Regulation model of TAR mediated by non-canonical IAA33. P: phosphate group; MPK14: MITOGEN-ACTIVATED PROTEIN KINASE 14; Green ARFs: Repressive ARFs; Pink ARFs: Activate ARFs (Adapted with permission from Ref. [16]. 2020, Lv); (**E**). Regulation model of TAR mediated by non-canonical IAA32/34. PM: plasma membrane; TMK1: transmembrane receptor kinase 1 (Adapted with permission from Ref. [60]. 2016, Cao). Arrows indicate promotion, and short lines indicate inhibition. AuxREs: auxin response elements.

**Figure 3 ijms-23-12495-f003:**
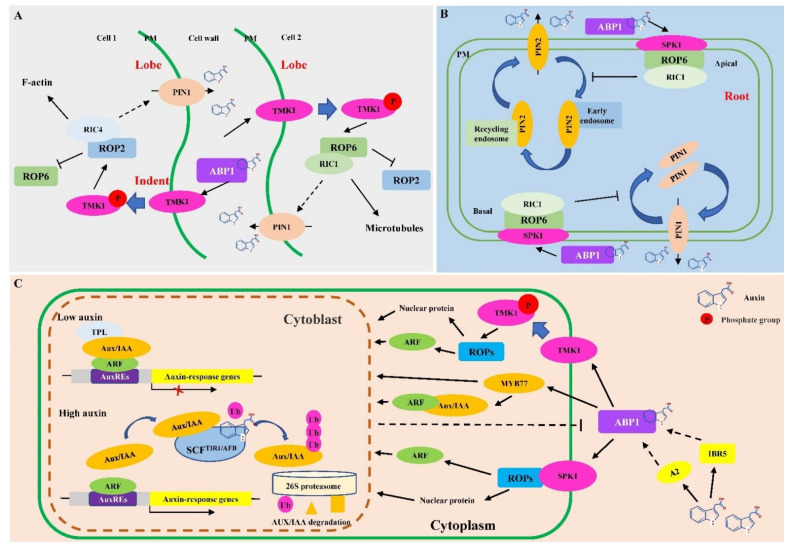
Regulation model of auxin response mediated by auxin binding protein 1. (**A**). Regulation model of non-transcriptional auxin response (non-TAR) mediated by ABP1-TMK1-ROPs. PM: plasma membrane; TMK1: transmembrane receptor kinase 1; P: phosphate group; ABP1: auxin binding protein 1; PIN1: PIN-FORMED 1; ROP2: Rho of plants-GTPases 2; ROP6: Rho of plants-GTPases 6; RIC1: CRIB motif-containing protein 1; RIC4: CRIB motif-containing protein 4 (Adapted with permission from Ref. [107]. 2014, Chen); (**B**). Regulation model of Non-TAR mediated by ABP1-SPK1-ROPs. PIN2: PIN-FORMED 2; SPK1: SPIKE1 (Adapted with permission from Ref. [105]. 2015, Feng); (**C**). Regulation model of transcriptional auxin response mediated by auxin binding protein 1. AUX/IAA: auxin/indole-3-acetic acid; ARF: auxin response transcription factor; TPL: TOPLESS; EAR: ethylene response factor-associated amphiphilic repression domains; PB1: Phox and Bem1; DBD: B3-type DNA binding domain; SCF^TIR1/AFB^: SCF (Skp-Cullin-F-box) ubiquitin ligase (E3) complex containing transport inhibitor response 1 protein (TIR1) or auxin signaling F-box protein (AFB); Ub: ubiquitin; AuxREs: auxin response elements; A2: phospholipase A2; IBR5: IBR5 protein phosphatase; Arrows indicate promotion, and short lines indicate inhibition.

**Table 1 ijms-23-12495-t001:** Effects of genetic modification of auxin response genes on plant traits.

Gene	Pathway of Auxin Response	Detail of the Modification Traits	Type of Regulation	Species	References
1. Changes of the apical dominance					
*OsIAA1*	*CTAR*	Decreased plant height, loose plant architecture, reduced inhibition of root elongation to auxin treatment	Overexpression	*Oryza sativa*	[140]
*SlIAA15*	CTAR	Reduced apical dominance, increased lateral root formation, and decreased fruit set	Down-regulated	*Solanum lycopersicum*	[141]
*ARF7/ARF19*	CTAR	Thin and short florescence stems, enhanced apical dominance, and reduced and delayed lateral root formation	Double mutant	*Arabidopsis*	[142,143]
*VvIAA19*	CTAR	Faster growth including root elongation and earlier floral transition	Overexpression	*Arabidopsis*	[144]
**2. Root architecture and growth**					
*EgrIAA4*	CTAR	Inhibition of primary root elongation, lateral root emergence and gravitropism	Overexpression	*Arabidopsis*	[145]
*SlIAA27*	CTAR	Root development alternations	Silencing	*Solanum lycopersicum*	[146]
*MtARF2*, *3*, and *4*	Non-TAR	Reduced primary and lateral root length, but increased lateral root density	Simultaneous knockdown	*Medicago truncatula*	[147]
**3. Stem characteristics modification**					
*PttIAA3*	CTAR	Changes of secondary xylem development	Overexpressing mutant gene	*Populus tremula × P. tremuloides*	[148]
*SlIAA9*	CTAR	Including enhanced hypocotyl/stem elongation, increased leaf vascularization, and reduced apical dominance	Silencing	*Solanum lycopersicum*	[149]
*GhARF2*	Non-TAR	Inhibited fiber cell elongation but promoted initiation	Overexpression	*Gossypium hirsutum*	[150]
**4. Modification of leaf traits**					
*CgARF1*	*CTAR*	Increased leaf senescence and decreased chlorophyll content.	Overexpression	*Arabidopsis* and *Nicotiana benthamiana*	[151]
*SlARF3/ETT*	TAR	Decreased density of epidermal pavement cells and reduced density of trichomes of leaves and shoot xylem cells	Silencing	*Solanum lycopersicum*	[152]
*e2fc*	TAR	Reductions in ploidy levels of mature leaves	Silencing	*Arabidopsis*	[80]
**5. Modification of flower and fruit traits**					
*RhIAA16*	CTAR	Promoted petal abscission	Silencing	*Rosa chinensis*	[153]
*RsIAA33*	TAR	Inhibited the reproductive growth and promoted taproot thickening and development	Silencing	*Raphanus sativus*	[154]
*MdIAA26*	CTAR	Promoted the accumulation of anthocyanin in calli	Overexpressing	Malus domestica	[155]
*FaARF4*	Non-TAR	Flowering earlier	Overexpression	*Fragaria vesca*	[156]
*SlARF4*	Non-TAR	Accumulated starch at early stages of fruit development and enhanced chlorophyll content and photochemical efficiency.	Silencing	*Solanum lycopersicum*	[157]
*SlARF5/MP*	TAR	Seedless fruits following emasculation	Silencing	*Solanum lycopersicum*	[158]
*OsAFB6*	CTAR	Promoted inflorescence meristem development and increased panicles size and grain yield	Overexpression	*Oryza sativa*	[20]
*SlARF6A*	CTAR	Increased chlorophyll contents, accumulated starch and soluble sugars, and inhibited fruit ripening and ethylene production	Overexpression	*Solanum lycopersicum*	[159]
*PTRE1*	CTAR	short siliques and arrested embryogenesis	Mutant	*Arabidopsis*	[160]
*BnaA9.ARF18*	Non-TAR	Increased seed weight and silique length	Mutant	*Brassica napus*	[161]
*FveARF8*	CTAR	Larger fruit	Mutant with *FveRGA1*	*Fragaria x ananassa*	[162]
*SlARF10*	Non-TAR	Improved accumulation of starch, fructose, and sucrose in fruit	Overexpression	*Solanum lycopersicum*	[163]
**6. Response to environmental stress**					
*MdIAA9*	CTAR	Increased tolerance to osmotic stresses.	Overexpression	*N. tabacum*	[22]
*OsIAA18*	CTAR	Enhanced salt and drought tolerance	Overexpression	*Oryza sativa*	[18]
*VvIAA18*	CTAR	Improved salt tolerance	Overexpression	*N. tabacum*	[164]
*MdIAA24*	CTAR	Increase drought resistance and cadmium tolerance.	Overexpression	*Malus domestica*	[21]
*OsIAA20*	TAR	Enhanced drought and salt tolerance, increased stomatal closure, and reduced the rate of water loss	Overexpression	*Oryza sativa*	[19]
*SlARF4*	Non-TAR	Enhanced resistance to water stress and rehydration ability	Knockdown	*Solanum lycopersicum*	[165]
*CsARF5/MP*	TAR	Enhanced the cold stress tolerance	Overexpression	*Cucumis sativus*	[166]
*ABP1*	Non-TAR	Changes of root and hypocotyl growth and bending, lateral root and leaf development, bolting, as well as response to heat stress	Gain-of-function alleles	*Arabidopsis*	[167]
*ABP1*	Non-TAR	Higher root slanting angles, longer hypocotyls, agravitropic roots and hypocotyls, aphototropic hypocotyls, and decreased apical dominance.	Heterozygous abp1/ABP1 insertion mutant	*Arabidopsis*	[124]
*ABP1*	Non-TAR	Severe retardation of leaf growth	Knockdown	*Arabidopsis* and *N. tabacum*	[101]
*ABP1*	Non-TAR	Changes in leaf growth rate or form, the cell size in certain regions of the leaf was altered	Overexpression	*Arabidopsis*	[168]
*TMK1*	Non-TAR	Hypocotyl elongation	Double mutant	*Arabidopsis*	[168]

## Data Availability

Not applicable.

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
