# Peer review of "Molecular Mechanisms of Diverse Auxin Responses during Plant Growth and Development"

_ijms, 2022, doi:10.3390/ijms232012495_

Round 1
Reviewer 1 Report
Dear colleges, I'm glad to say this manuscript seems very informative and accurate. I could see only a few spelling mistakes there and the pictures are too miniatures for my opinion and need to be enlarged at least to 125%. The Table 1 needs small improvement in "Detail of the modification traits" - details for every modification have to be in one style beginning with capital letter.
In general it was a pleasure.
Author Response
Thank you very much for your great comments. We strongly agreed with your opinion. So, we revised the Figures to improve the quality. At the same time, we also revised the Table 1. Finally, each of our authors revised the manuscript again.

Reviewer 2 Report
Author has systematically summarized and classified the molecular mechanisms of diverse auxin responses, and comprehensively elaborated the characteristics and multilevel regulation mechanisms of the canonical transcriptional auxin response. Considering in depth coverage under the review, this study could useful to many researcher and readers for their further investigations.
Figures quality need improvement
i recommend this review can be considered into this journal after improvement of figure quality by authors and minor language checks
Author Response
Thanks very much for your comments. According to your suggestion, we have revised the manuscript and figure quality.
Reviewer 3 Report
Drae Authors,
It is an interesting review. I did some minor comments, however as it is not my main field of study (plant nutrition) it is better to refer to review results of other experts.
To my knowledge if you make a section for Auxin interactions with other hormones and effectors, it would be much more informative.

Author Response
Your comment: It is an interesting review. I did some minor comments, however as it is not my main field of study (plant nutrition) it is better to refer to review results of other experts.
Response: Thanks very much for your comments. Following your advice, we have revised the manuscript.
Your comment: To my knowledge if you make a section for Auxin interactions with other hormones and effectors, it would be much more informative.
Response: Thank you very much for your great comments. We strongly agreed with your opinion. This section for Auxin interactions with other hormones and effectors was also addressed in the early time when we prepared the manuscript. However, in the later revision process, we deleted this part of the content. Because we found that, on the one hand, the relationship between IAA and any one of other hormones is very complex due to a very large body of information, which cannot be clearly described in one or a few paragraphs. On the other hand, Excessive or enormous description of interaction of IAA and other hormones deviate from the core content of what we are currently focused on. We’d like to address this specifically in a separate review paper later. Anyway, your comments are very meaningful for our future research, and we will pay attention to the relationship between IAA and other hormones, and do this later. Thank you very much.